# Quantifying Causality between Climate Change and Credit Risk: A Bibliometric Study and Research Agenda

**Noluthando Mngadi** *,† and **Hossana Twinomurinzi** †

Centre of Applied Data Science, University of Johannesburg, Johannesburg 2006, South Africa; hossanat@uj.ac.za
* Correspondence: noluthando.mngadi@gmail.com
† These authors contributed equally to this work.

**Abstract:** There is increasing pressure on organisations and countries to manage the financial risks associated with climate change. This paper summarises research on climate change, credit risk and the associated losses, and specifically identifies methods that could contribute to quantifying the causal relationships between climate change and credit risk. We conducted a bibliometric analysis using the Web of Science database to analyse 3138 documents that investigated climate change and credit risk. The key results reveal that climate change has a quantifiable effect on credit risk, and that the most affected industries or sectors are energy, transportation/mobility, agriculture and food, manufacturing, and construction. The prominent methods to quantify causal relationships between climate change and credit risk are regression models, but these are mostly used in preliminary and testing stages. Distance to default and credit risk are the main areas of focus when quantifying climate change and credit risk. Banks are the main type of organisation that have sought to quantify the causal relationship. We identify a research agenda to quantify these causal relationships.

**Keywords:** credit risk; climate risk; climate change; modelling; quantification; default; probability; bibliometrics

## 1. Introduction

There is an increasing need to identify, quantify, and mitigate the financial risks associated with climate change. The potential impact of climate change on economics, industries and individual increases, financial industry participants are shifting focus towards development of robust and credible methodologies for quantifying the impact of climate risk on their operations, and by extension, the operations of their clients.

Climate change is defined as a long-term change in the average weather patterns that have come to define the earth's local, regional and global climates [1]. As these events are becoming more frequent and more severe. Governments is increasingly seeking to reduce the effects of climate change and global warming, especially in the form of carbon pricing mechanisms, consumers and organisations are increasingly concerned about the implications of climate change on their financial assets [2,3]. Organisations and governments are similarly exploring methods for quantifying these climate risks.

Climate change is one of the greatest global concerns, posing unprecedented challenges to the governance of global, socioeconomic and financial systems. Global warming is the biggest contributing factor to climate change, which is being measured by the gradual increase in the overall temperature of the earth's atmosphere. This is generally attributed to the greenhouse effect, which is caused by increased levels of carbon dioxide, chlorofluorocarbons (CFCs), and other pollutants in the atmosphere.

The effects of climate change can occur through physical risks and transition risks. Physical risks arise from the effect of intense climate events such as hurricanes, floods, or droughts; transition risks are the end result of changing policies, practices, and technology as companies move towards a low-carbon economy. Physical risks and transition risks

are not independent of each other. Efforts to minimise global warming effects can reduce physical risks. However, an increase in transition risks can arise through higher market prices, disruptive technologies, and increasing regulatory costs. Credit risk is defined as the risk of financial loss due to the borrower failing to repay interest and some or all of the principal on the loan [4]. Climate-related risks affect all aspects of credit risk, that is, the borrower's ability to make earnings to service and repay the debt, as well as the capital and collateral to support the loan.

The interest on embedding climate change risks to credit risk has spiked quite significantly from the years of 2015. This sudden interest was inspired by the 2015 Paris Agreement, where 195 countries pledged to limit global warming levels by the year 2050 [5]. Governments are now, therefore, under increasing pressure to be more stringent on carbon pricing mechanisms to limit emissions. Financial institutions are also impacted by this agreement, as they need to consider asset pricing mechanisms that might be affected by the transition policies to environmental friendly processes. In addition, the after effects of extreme climate change events impacts the asset values. One of the challenges in quantifying climate-related risks is measuring them in order that they may be incorporate into quantification methods.

Therefore, the main objectives of this study were to synthesise the research on the impact of climate change on expected credit risk losses, to identify quantification methods and to propose directions for future research in the field. Specifically, we sought to answer the following questions: (i) What is the long term impact of climate change on credit risk? (ii) If there is an impact, which credit risk products are most affected? (iii) Which industries or sectors are most affected by the climate change risks? (iv) What are the currently available methodologies to quantify climate change on credit losses? To answer these questions, a bibliometric analysis with a focused systematic literature review was adopted as the most appropriate method.

The key findings show that the study of climate change and credit risk dates back to the 1990s with an annual publication rate of 11.94%. The United States of America (USA) is the country with the highest number of publications followed by China. The Journal of Geophysical Research Atmospheres, which originates in the USA, is the most local cited source from the reference list. The greatest financial loss from climate change is the Texas, USA Hurricane Harvey in 2017 estimated at USD 125 billion [6]. The most affected sectors are energy, transportation or mobility, agriculture and food, manufacturing, construction, and mining. Organisations in various industries are looking for ways to decrease their emissions, water usage, and environmental impacts in general. The factors used to quantify climate change risks are carbon emissions, extreme weather events such as draughts, floods, extreme temperatures and climate change scores. Credit risk is quantified by distance to default, probability of default, loan (debt) amount, and asset refinancing. Some of the quantification models that are used to incorporate climate change risks to credit risk losses are: logit, quantile regression, support vector machines, random forests, EZ-climate models and index-based modeling. The greatest challenge is the availability of data to quantify climate-related risks. This study contributes to the literature by synthesising the literature on the topic, highlighting the risks associated with climate change, and providing the research agenda and the future work.

The remainder of this paper is structured as follows: Section 2 presents the research method, Section 3 presents the results of the bibliometric analysis and focused systematic literature review, and the discussion of the results. Section 4 comprises of the conclusion, provides the main implications of the study and makes suggestions for further research on climate change and credit risk.

## 2. Research Method

### 2.1. Bibliometric Analysis

Bibliometric analysis was first applied in 1969 by Pritchard [7] and has been widely used in academic research. It is commonly used for investigating and analysing large

amounts of scientific data [8]. In general, research on the impact of climate change focusing specifically on the analysis of credit risk has been conducted independently of other climate change-related topics [9–11]. We used the R software package, Bibliometrix, developed by Aria and Cuccurullo [12], to analyse and visualise the data or information from the data mining of the literature. The R software was used because it offers the bibliometrix package, which has all the tools to complete bibliometric analysis. Vosviewer was used due to its ability to easily interpret relationships between publications in a graphical format [13]. It is another version of bibliometric software that we used to visualise maps or networks. The data mining approach applied in this study was divided into four main steps (as per Figure 1): (i) select the database to query, namely, the Web of Science (WoS), (ii) define the search string, (iii) query the database using the defined search string, and (iv) select the relevant documents (articles, conference proceedings, and reviews) that have undergone the peer review process to ensure quality.

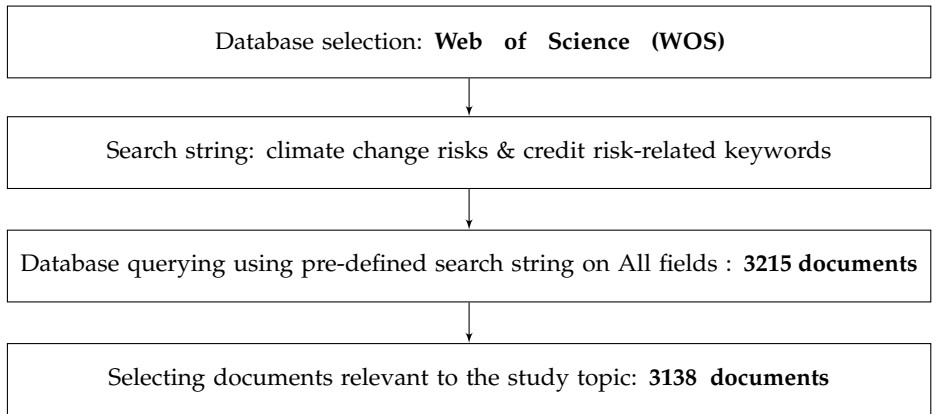

**Figure 1.** Summary of the data mining approach.

The string that was used to query or search the full records of all the documents in the WoS database was:

("credit" or "credit risk*" or "credit default" or "default risk" or "default") and ("climate" or "climate change" or "climate risk*" or "weather changes" or "global heating").

The search string was run on 12 May 2022 and gave an output, after applying the inclusion and exclusion criteria, of 3138 documents.

The inclusion criteria led to a selection of documents in the English language only and document types labelled as articles, reviews, book chapters, and conference proceedings. The selected documents had had undergone a peer review process prior to their publication thus ensuring their quality [14]. Documents that were less likely to have undergone a peer review process such as letters and unpublished material were excluded from the final dataset used for the bibliometric analysis.

*2.2. Systematic Review*

A systematic review makes use of studies to collect secondary data and existing evidence to systematically analyse the information [15].

### 2.2.1. Identification of the Articles

The articles were identified by using the findings from the bibliometric analysis outlined in Section 2.1. The search string was run on the most cited sources (journals) and the inclusion and exclusion criteria were applied.

### 2.2.2. Construction of the Search Terms

The study used the search string applied in the bibliometric analysis to search for relevant articles that would assist in answering the research questions.

### 2.2.3. Identification of the Data Sources

The search string was first run on the Scopus database, but this yielded less than 20 papers. The same search string was then run on the WoS database where it yielded more than 3000 documents.

### 2.2.4. Inclusion and Exclusion Criteria

The queried search string yielded 3138 documents that included articles and journals. Documents in the form of published articles and proceedingśpapers that employed quantification methods to model climate change effects on credit risk were then selected. This resulted in 12 papers for systematic review.

### 2.2.5. Structured Classification and Coding Framework

The study formulated categories that were used to effectively analyse and classify the body of information from the systematically selected papers. The classification framework employed letters and numbers to code the articles as per Table 1.

**Table 1.** Classification framework and the distribution of articles per category.

| Category | Level 1 | Code | No. of Articles |
|---|---|---|---|
| Climate focus | Adverse weather | A1 | 3 |
| | Climate change | A2 | 6 |
| | Greenhouse emissions | A3 | 6 |
| Credit focus | Transition risk | B1 | 2 |
| | Index insurance | B2 | 1 |
| | Distance to default | B3 | 5 |
| | Access to credit (finance) | B4 | 2 |
| | Credit rating | B5 | 2 |
| | Cost of debt | B6 | 2 |
| | Risk (credit risk) | B7 | 7 |
| Credit organisation | Microfinance | C1 | 1 |
| | Banks | C2 | 5 |
| | Bond markets | C3 | 2 |
| Sector | Agriculture | D1 | 4 |
| | Non-financial enterprises | D2 | 2 |
| | Organisations (firms) | D3 | 7 |
| Modeling methods | Regression (supervised) | E1 | 9 |
| | Classification (supervised) | E2 | 3 |
| | Unsupervised | E3 | 2 |
| Types of data | Credit data | F1 | 10 |
| | Panel data | F2 | 10 |

## 3. Analysis and Discussion of Findings

In this section, we examine and discuss the findings from the bibliometric analysis and systematic review.

### 3.1. Bibliometric Results

The summary of the final dataset used in the bibliometric analysis is in Table 2. The table provides a summary of the main information including the timespan, document

citations, document types, references, document contents, authors' information including collaborations. The documents were published between 1991 and 2022, were from 1018 sources, had 11,606 authors, 136,661 references, and a collaboration index of 3.98.

**Table 2.** Main information about the dataset.

| Description | Results |
|---|---|
| Timespan | 1991:2022 |
| Sources (Journals, Books, etc.) | 1018 |
| Documents | 3138 |
| Average years from publication | 6.03 |
| Average citations per year per doc | 3.012 |
| References | 136,661 |
| Document types | |
| Article | 2863 |
| Article; proceedings paper | 49 |
| Document contents | |
| Keywords plus (id) | 6752 |
| Author's keywords (de) | 8053 |
| Authors | |
| Authors | 11,606 |
| Authors collaboration | |
| Single-authored documents | 293 |
| Documents per author | 0.27 |
| Authors per document | 3.7 |
| Co-Authors per documents | 4.55 |
| Collaboration index | 3.98 |

The scientific publications on climate change and credit risk began in 1991 with an annual an publication increase rate of 11.94%. The number of papers published yearly is plotted in Figure 2. A clear upward trend of publications can be seen during the sample period. Noticeably, as opposed to the slow and steady growth trend before 2015, the number of relevant publications increased sharply since 2015 (the year of the aforementioned Paris Agreement), which indicates a substantial increase in interest on the part of academics in climate change and credit risk. The number of publications in year 2022 was lower because it is not full year; the string was run on 12 May 2022; 12 May represents 131 of 365 days.

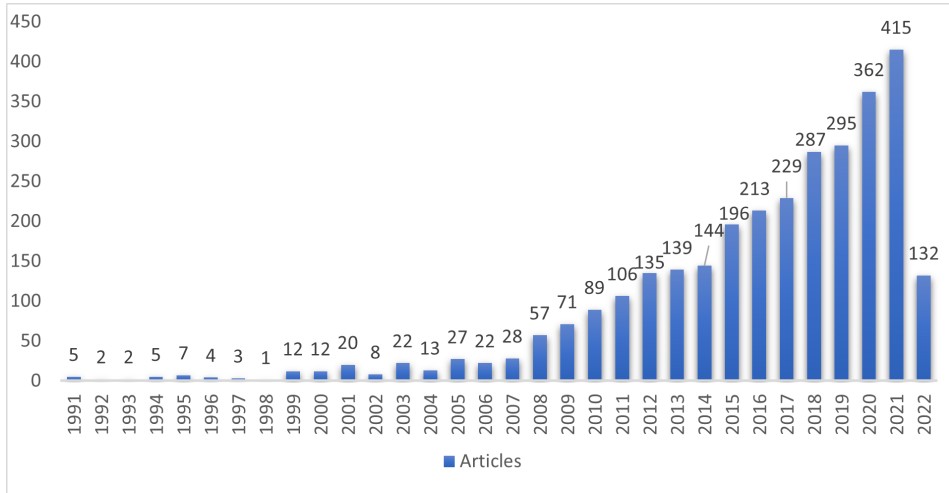

**Figure 2.** Number of relevant publications on climate change and credit risk per year.

Table 3 provides a list of the top 20 institutions with the highest number of publications or articles. The Nanjing University of Information Science and National Center for Atmospheric Research are the leading institutions with 63 and 59 articles, respectively,

followed by the University of California with 54 articles. These institutions are from China and the USA, and these countries have high production outputs and citations, as reflected in Table 4.

**Table 3.** Top 20 institutions by number of articles.

| Rank | Affiliations | Origin Country | Articles |
|------|-------------|----------------|----------|
| 1 | Nanjing University of Information Sc | China | 63 |
| 2 | National Center for Atmospheric Res | UK | 59 |
| 3 | University of California | USA | 54 |
| 4 | University of Chinese Academy Sc | China | 51 |
| 5 | University of Colorado Boulder | USA | 50 |
| 6 | Institute of Atmospheric Physics | China | 48 |
| 7 | Colorado State University | USA | 44 |
| 8 | University of Maryland | USA | 42 |
| 9 | Wageningen University Res | Netherlands | 42 |
| 10 | Sun Yat-sen University | China | 40 |
| 11 | Purdue University | USA | 39 |
| 12 | Nanjing University | China | 34 |
| 13 | University of Cambridge | UK | 34 |
| 14 | University of Florida | USA | 34 |
| 15 | Tsinghua University | China | 32 |
| 16 | University of Michigan | USA | 32 |
| 17 | The University of Wisconsin | USA | 32 |
| 18 | University of Oxford | UK | 31 |
| 19 | University of Toronto | Canada | 31 |
| 20 | The Swiss Federal Institute Tech | Switzerland | 30 |

**Table 4.** Top 20 Countries with highest number of publications.

| Rank | Country | Prod Freq | Total Cit |
|------|---------|-----------|-----------|
| 1 | USA | 2402 | 25,324 |
| 2 | CHINA | 1121 | 5019 |
| 3 | UK | 577 | 4584 |
| 4 | GERMANY | 571 | 3892 |
| 5 | CANADA | 441 | 2870 |
| 6 | AUSTRALIA | 429 | 4855 |
| 7 | FRANCE | 317 | 1662 |
| 8 | NETHERLANDS | 263 | 2031 |
| 9 | INDIA | 241 | 1604 |
| 10 | ITALY | 228 | 1274 |
| 11 | BRAZIL | 207 | 823 |
| 12 | JAPAN | 180 | 1052 |
| 13 | SOUTH KOREA | 173 | 420 |
| 14 | SPAIN | 165 | 1110 |
| 15 | SWEDEN | 152 | 955 |
| 16 | SWITZERLAND | 151 | 1452 |
| 17 | SOUTH AFRICA | 124 | 1397 |
| 18 | BELGIUM | 121 | 853 |
| 19 | KENYA | 120 | 543 |
| 20 | PAKISTAN | 110 | 514 |

Table 4 shows the top 20 countries contributing to research on climate change and credit risk. The USA is the leading country in this regard with 2402 publications, followed by China with 1121. The USA has thus produced almost double the number of publications than China. Publications from the USA also attract a high number of citations, 5.5 times more than that of China. The World Bank, the United Nations (UN), and many other influential international organisations which are focused on climate change are headquartered in the USA organisations are likely to facilitate and encourage research in the topic area.

Figure 3 lists the most frequently used words based on the authors keywords and keywords plus. "Climate change", "impact" and "model" are the top three keywords used by authors and keywords plus. "Policy", "risk" and "carbon tax" are among the used keywords and this clearly indicates the relation to policy around climate change financing and the risks associated. Climate change seems to have attracted much attention.

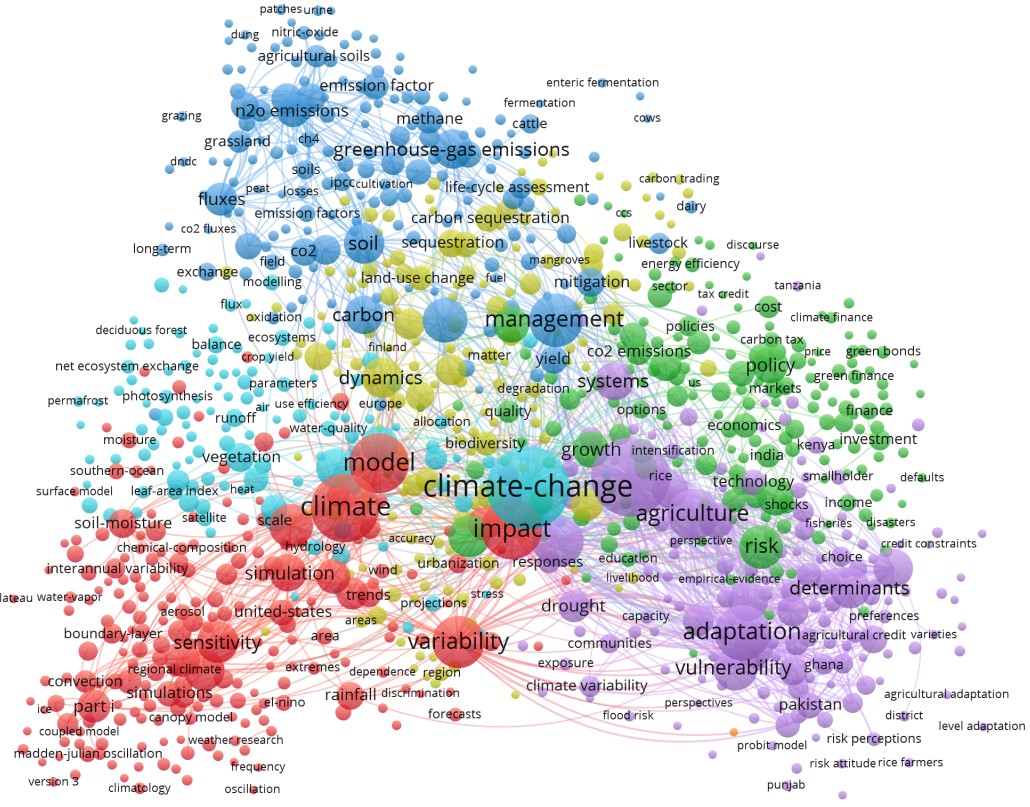

**Figure 3.** Most frequently used keywords by authors.

The country co-authorship network on Figure 4, shows the interrelation between countries. From the figure, the USA is at the heart or center of network, which is not surprising as the USA has the highest number of publications with 2402 publications and highest citations with 25,324 total citations. It is also noticeable that there is regional collaboration with the US and Europe working together, while the the other regions work together as continents.

The most local cited sources on Table 5, shows the journals with the most cited number of articles. These journals have a common scope, which is the science and the environmental affairs.

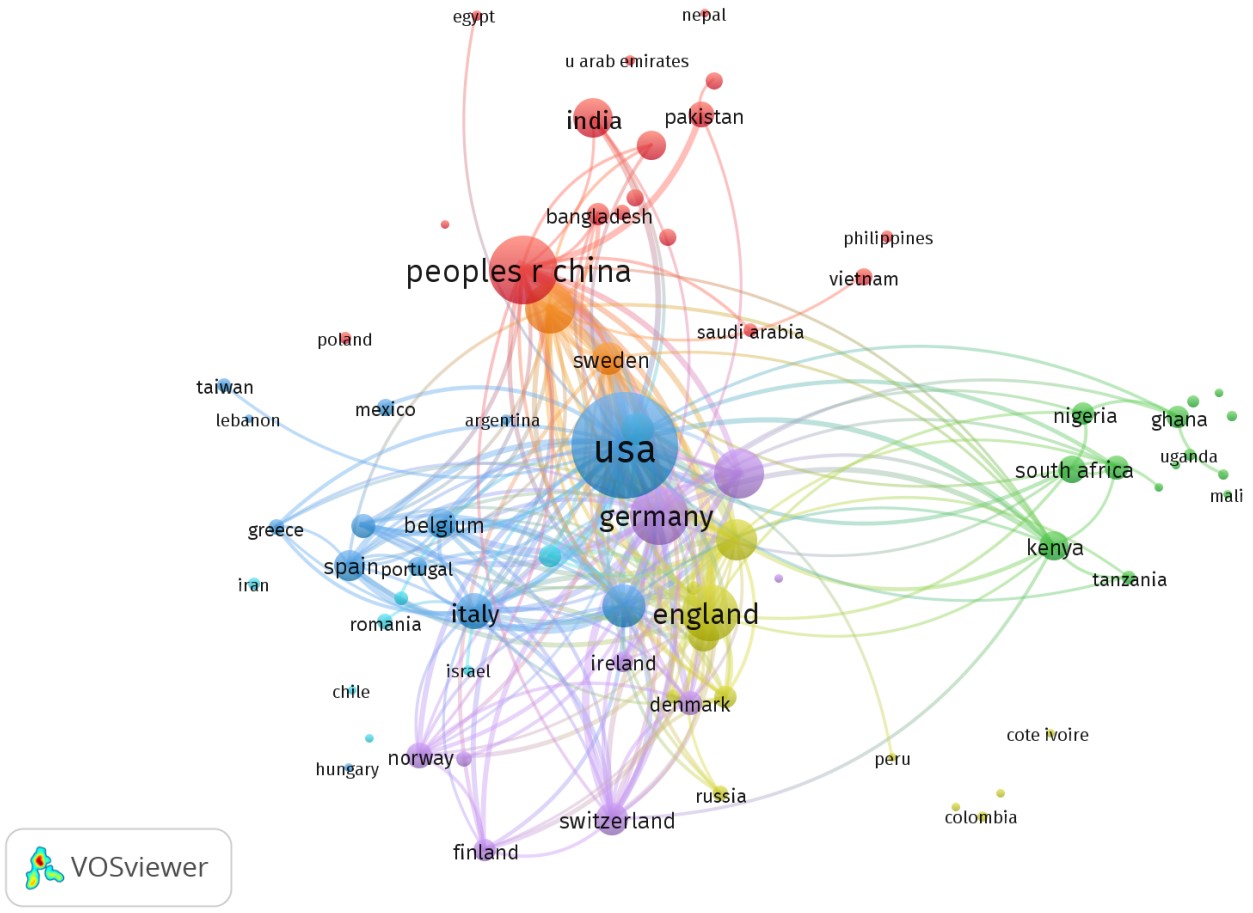

**Figure 4.** Country co-authorship network.

**Table 5.** Most local cited sources (from reference lists).

| Rank | Sources | Journal Scope | Articles |
|---|---|---|---|
| 1 | Journal of Geophysical Research-Atmospheres | Atmospheric properties and processes | 3879 |
| 2 | Journal of Climate | Dynamics and physics of the climate system | 2430 |
| 3 | Atmospheric Chemistry and Physics | Earth's atmosphere and the underlying chemical and physical processes. | 2041 |
| 4 | Science | Scientific discovery | 1862 |
| 5 | Geophysical Research Letters | Major geoscience disciplines | 1671 |
| 6 | Global Change Biology | Biological systems and environmental change | 1539 |
| 7 | Nature | Natural science and technology | 1490 |
| 8 | Monthly Weather Review | Atmospheric circulations and physics | 1419 |
| 9 | Proceedings of the NASA | Biological, physical, and social sciences | 1312 |
| 10 | Climatic Change | Climate change and variability | 1194 |
| 11 | Journal of Hydrology | Hydrological sciences | 1166 |
| 12 | Journal of the Atmospheric Sciences | Physics, dynamics, and chemistry of the atmosphere | 1162 |
| 13 | Global Environmental | Change Human and policy dimensions of global environmental change | 1089 |
| 14 | Water Resources Research | Hydrology, water resources, and the social sciences of water. | 1083 |

**Table 5.** *Cont.*

| Rank | Sources | Journal Scope | Articles |
|------|---------|---------------|----------|
| 15 | Energy Policy | Political, Economic, Planning, Environmental and Social Aspects of Energy. | 1072 |
| 16 | Agriculture, Ecosystems & Environment | Agroecological science | 1036 |
| 17 | Atmospheric Environment | Atmospheric composition and its impacts. | 1030 |
| 18 | Agricultural and Forest Meteorology | Inter-relationship between meteorology, agriculture, forestry, and natural ecosystems. | 949 |
| 19 | Climate Dynamics | Dynamics of the global climate system. | 948 |
| 20 | Bulletin of the American Meteorological Society | Significance for the weather, water, and climate community | 929 |

*3.2. Systematic Review*

This paper places a special emphasis on quantitatively measuring causality between climate change and credit risk. We therefore selected those papers which employed quantitative methods to measure the causality. Tables 1 and 6 show a summary of the papers and a classification method to analyse the papers.

The classification groups were created by categorising articles according to their focus or study subject. The first category is climate focus where the majority of the papers quantified causality of climate change and emission rates against credit risk. Second is the credit focus, where the majority of articles examined credit risk factors, followed by the distance to default. Transition risk, index insurance, access to credit, credit rating, and cost of debt credit were not examined much. Third is the credit organisation which offers financial services to those who require them. Banks are leaders in the financial service providers, followed by bond markets and microfinance. Fourth is the sector on which the articles focused, and organisations (firms) comprised the majority sector, followed by the agricultural sector and non-financial enterprises. The fifth category comprises the quantitative modelling methods that were adopted by each study, and the majority of papers in this category applied supervised regression models followed by classification and unsupervised models. This is no surprise, as traditionally, credit risk modelling employs regression methods. The last classification category was the type of data that was applied in the study. Findings show that panel data and credit data were equally used. This validates our systematic method of selection, whereby the inclusion and exclusion criteria involved selecting papers that quantify climate change risks together with credit risk.

The threat of climate change to financial stability may be felt in various ways. Extreme weather and other natural disasters, such as floods and droughts, are causing financial assets to lose value. Asset price adjustments in the transition to a low-carbon economy may pose transition hazards. The International Energy Agency (IEA) and an increasing number of nations are taking steps to reduce greenhouse gas emissions and encourage private investment in cleaner alternatives [16]. As a result, all carbon-emitting assets [17] may be liable to regulatory penalties, affecting investors' perceptions of future profitability, company sustainability, and creditworthiness. Unexpected and rapid regulatory changes might lead to a fire sale, precipitating a financial catastrophe [18].

**Table 6.** Summary of each paper's key focus and findings.

| Ref # | Author(s) Name [Framework or Model] | Key Focus and Findings of the Paper |
|---|---|---|
| 1 | Capasso, Gianfrate, Spinelli (2020) [regression] | This study investigated the link between climate change vulnerability and corporate credit risk. The key findings show that central banks should be more worried with climate risks, that the ability to get loans and bonds is negatively impacted by exposure to climate-related risks, and that financial regulators and policymakers should carefully evaluate the impact of climate change concerns on the stability of lending intermediaries and corporate bond markets. The key challenges are that industries in Scope 1 (direct) Scope 2 (indirect) and Scope 3 (supply-chain related) carbon emissions should ideally also be taken into account when assessing carbon emissions. The other key challenge is in methodologically assessing and estimating carbon emissions in present times [2]. |
| 2 | Kaur Brar, Kornprobst, Braun, Davison, Hare (2021) [framework] | This research quantified the impact of rising temperatures on credit risk. Increasing temperature is one metric of climate change. The researchers provided a framework for the methods that can be applied to the challenge of optimising a portfolio of agricultural loans made to corn farmers across a variety of corn producing regions in Ontario, Canada, while taking into account a number of potential climate change outcomes. According to the findings of the study, the loan portfolio that was obtained by taking into consideration climate change is associated with a lower risk compared to the lending portfolio that was obtained by ignoring climate change. This study did not consider the possibility of more extreme weather patterns as a result of climate change, so there's plenty of room for future research [19]. |
| 3 | Birindelli, Bonanno, Dell'Atti, Iannuzzi (2022) [fixed effects] | This paper investigated whether or not financial institutions that have a greater commitment to addressing climate change issues, as measured by the climate change score provided by the Carbon Disclosure Project (CDP), experience an impact on their credit risk. According to the findings, when banks pay a medium-to-high degree of attention to climate concerns, the amount of risk associated with bank loans decreases, suggesting that the dedication to climate issues is responsible for this effect [20]. |
| 4 | Möllmann, Buchholz, Kölle, Musshoff (2020) [regression] | The researchers used a unique borrower dataset given by a Microfinance Institution (MFI) to examine whether remotely sensed plant health indicators may explain the MFI's agricultural loan portfolio credit risk in Madagascar. The results show that the MFI's credit risk is largely accounted for by the vegetative health metrics used. The explanatory value of the vegetative health indicators rises with rising credit risk, as shown by quantile regressions, as well. The future study should focus on developing meso-level vegetation-based index insurance products. Crop-specific meso-level index insurance products may be useful for smallholder farmers in research region because of the prominence of rice farming. It would be beneficial to focus on using remote-sensed data with a greater resolution in order to boost accuracy [21]. |
| 5 | Shobande (2019) [time series] | Using a time-series technique, this study assessed the impact of monetary policy on climate change in the East African Community. The study also examined the short- and long-term climate change effects of monetary policy. According to the study's empirical results, monetary policy can ease the transition to low-carbon economies through the credit and interest rate channel, but at the cost of financial instability. The findings also show that a lack of funding for the environment might lead to an increase in $CO_2$ emissions and a decrease in biodiversity. As a result of the effects of climate change, the economy is likely to decline even further. Climate funding and stock market volatility can be examined in future study, which may add value to current findings [22]. |
| 6 | Yu, Li, Mirza, Umar (2022) [machine learning] | Using a variety of machine learning algorithms, this study predicted the credit ratings of environmentally friendly companies. The analysis of the study shows that machine learning approaches such as classification and regression trees were most accurate in predicting credit ratings. Accuracy was maintained even when predicting was limited to investment, speculative, or default categories. Additionally, the results show that a random forest ensemble may be used to forecast default or near-default ratings in conjunction with regression trees [23]. |

**Table 6.** *Cont.*

| Ref # | Author(s) Name [Framework or Model] | Key Focus and Findings of the Paper |
|---|---|---|
| 7 | Javadi, Masum (2021) [regression] | This study determined if and how much climate risk influences the cost of financing for businesses. It established if banks regard global warming as an important risk factor and whether they include it in various aspects of their loan agreements. The analysis in the research shows that businesses in areas with a greater risk of climate change pay much higher interest rates on their bank loans. Lenders are increasingly considering climate change to be a relevant risk factor, according to historical data. The study mainly analyzes the long-term loans and their adverse effects due to climate change, adding short-terms loan with respect to customers exposure to climate risk and its impacts on a firms cost of borrowing can be considered as future research [24]. |
| 8 | Agliardi (year 2022) [framework] | This paper focused on analyzing green securization strategy's impact on bank portfolio exposure and alignment with global climate objectives using a unique model. The result of the study shows, the applied technique of green securization has positive impact on the climate risk exposure of financial institutions and their alignment with global climate objectives. It is evident that, additional empirical research is needed to offer credible data for the study to predict properly what would happen if the quantity of these financial instruments explodes [25]. |
| 9 | Kabira, S.Rahmana, Md.Rahmana, Anwar (2021) [regression] | This study looked at the the impact of carbon emissions on the likelihood of a company's bankruptcy. It examined the influence of emissions on default risk and the economic channels via which they affect the performance and value of a company. The findings show that carbon emissions have a significant influence on default risk. Distance-to-default, a reversible indicator of default risk, is significantly impacted negatively by carbon emissions. In addition, the study found that carbon mitigation techniques at the corporate level can assist reduce the risk of default [26]. |
| 10 | Hogersthal, Lui, Tomicic, Vidovic (2019) [model market signals] | Researchers in this study presented a valuation-based technique to predict how the risk of an energy transition might affect the creditworthiness of publicly traded corporations throughout the world in the next 30 years. The research finding's indicate that, utilities, minerals, energy, and consumer staples might be the most default-prone businesses during a rapid transition. Over the next 30 years, there is a risk that some large-revenue corporations from these industries would default on their financial commitments, which could have significant rippling effects on the economy, both at an international and national level. The data used for analysis in this research is based on one specific region, if the sample of data is enhanced globally, the result will be more accurate and precise [27]. |
| 11 | Garbarinoa, Guina (2021) [regression] | The research focused on repeating mortgage and property transactions that occurred surrounding a catastrophic flooding disaster that took place in England in 2013–2014. The results show that firstly, lender values do not take local price decreases into account. As a result, the value of a company is artificially inflated. Second, lenders do not modify interest rates or loan amounts to compensate for this valuation bias. It is no coincidence that low-risk buyers tend to go into high-risk flood zones [28]. |
| 12 | Georgopoulou et al (2013) [framework] | This research presented a methodology and a decision-support tool for calculating the monetary risks that banks face as a result of the vulnerability of their loan recipients and/or applicants to climate change. The results show that the climate change risks for banks are substantial, and hence decision-makers need to quantify the extent of these risks and maybe incorporate them into the credit management process and environmental planning processes [29]. |

Agricultural productivity can be negatively affected by extreme weather conditions, putting smallholder farmers at risk of failing to repay their debts on time [21]. Vanelli and Kobiyama [30] found that a bank's vulnerability to high default rates is heightened by natural catastrophes, but that the bank's exposure may be reduced by assigning more green loans to its overall loan portfolio [20]. People, businesses, towns, and other organisations are all affected by the implications of climate change on financial risk because of physical dangers that destroy actual assets such as infrastructure and property. The increasing frequency of harsh weather can lead to losses or damage to assets, devaluation of land and homes and higher default risks in the mortgage portfolio, for example, extreme weather

may also affect agricultural productivity and cause farmers to fall behind on their loan payments if the amount they owe exceeds their farm revenue [31].

Previous studies suggest that climate-related risks such as extreme weather events, draughts, floods, etc., impact the stability of lending intermediaries and corporate bond markets [2]. Further, exposure to the climate-related risks negatively affects the ability of small business owners, farmers and retail customers to obtain loans and bonds [24,28]. This highlights the importance of financial regulators and policymakers to begin to evaluate and take into account the impact of climate changes on credit risk [29]. The analysis has shown that the optimization of a loan portfolio should take account of temperature and climatic change outcomes. The fact that climate change is ignored increases credit risk, but this study acknowledges the need for further research to investigate impacts of more severe natural weather patterns due to climate change [19]. The link between a financial institution's commitment to addressing climate change and credit risks is analysed on this study. These findings suggest that the associated risk of borrowing is reduced as a result of increased attention to climate issues. Financial institutions which have medium to high climate change commitments experience lower credit risks and emphasise the importance of earmarking resources for issues related to climate change [20,21]. Other results show that during the transition to a rapidly evolving energy source, industry sectors such as utilities, minerals, energies and consumer staples are more vulnerable to default. The analysis highlights the potential consequences of failure by large revenue corporations to fulfil their financial obligations and calls for more accurate estimates based on global data [27].

It is essential to understand the factors that are used to quantify climate change risks and credit risks. A majority of the studies employ the following proxies to quantify climate change: Möllmann [21] used remotely sensed vegetation health indices which are vegetation condition index, temperature condition index, vegetation health index to explain climate risk. Other studies employed carbon emissions such as [2,21,32,33], draughts [24,34], floods [28,34], extreme weather and extreme temperatures [20,34], climate change score [20]; to quantify the impacts of climate change on credit risk. Credit risk is widely quantified by using the measures such as distance to default, loan (debt) amount, mortgage refinancing, credit ratings, probability of default [2,19,23,26].

There are at least five quantitative approaches that the reviewed articles applied to model the effects of climate change on credit risk assessment and losses. For example, (i) Ref. [21] used sequential logit models and quantile regressions to investigate the explanatory power of remotely sensed vegetation health indices for credit risk at the aggregated bank level, (ii) Ref. [23] employed support vector machine, random forests ensemble, and artificial neural networks to forecast the credit ratings of eco-friendly firms; and (iii) Ref. [28] applied the logistic regression to examine how lenders account for extreme weather by comparing the mortgage transactions before and after severe floods. These approaches have interesting implications that can be taken into consideration when assessing credit risk.

Climate change events may reduce the value of properties affected, disrupt the price and mortgage rates of neighboring properties, increase borrowers' default rates, and worsen banks recovery rates from collateral fundamental underlying loans. One of the primary policy tools considered by governments to facilitate the transition to a low-carbon economy is the implementation of a carbon tax that penalises firms that emit greenhouse gases. Companies in high-emission sectors, such as energy (coal, oil, and gas), airlines, and steel manufacturing, may face higher operating costs or significant shifts in asset values, depending on the steepness of the curve, the new (or incremental) taxation regime, as well as their rate of adoption of green technology and/or making use of renewable energy sources.

## 4. Conclusions, Limitations and Future Work

Climate change does not only pose a serious threat to life on our planet, but it also has negative consequences for the global economy and the stability of financial systems through physical and transition climate risks. This paper found that the financial risks

associated with climate change can indeed be quantified, and this can result in better credit scoring regimes.

The impact of climate-related risk factors varies by severity and timing, the expected climate risks, the direct and indirect impacts on loan portfolios of borrowers and lenders, and the term of loan portfolios. The risks associated with climate change will affect each portfolio independently depending on many factors such as the location of the business, the industry/sector and the term of the credit facility, that is, why it is crucial for financial institutions to find methods to measure and quantify climate-related losses.

Effective management of credit risk is reliant on accurate estimation of expected credit losses which, in turn, relies on accurate estimation of the probability that a customer will default on a loan and expected recoveries post the point of default, including recoveries from sale of collateral.

Both the ability of a customer to generate cashflows and hence service debt, and the future value of loan collateral, can be impacted by climate change. Accurate estimate of expected credit losses therefore requires accurate modelling of the impact of climate change on associated risk measures.

### 4.1. Limitations

- This study analysed the documents that were mined (matched) by the search string. Other papers related to this study could have been missed because they used synonyms that were not included on the search string.
- Other studies were excluded because they were not applying or employing any quantitative methods, but they are also exploring the same topic.

### 4.2. Research Agenda and Future Work

Composite climate scoring for Multiple climatic events: The studies did not consider the possibility of more than one extreme weather event as a result of climate change. Multiple climate change factors, e.g., extreme rain, wind, temperature, floods, etc., can occur at the same time. We therefore consider the following question for further research:

- What are the factors that are used to measure climate change?
- How can multiple climate events be used to create a composite climate score?
- What are the different composite scores for different sectors?

Retail side of credit risk: Most of the studies focused on the corporate side of credit risk, but did not pay attention to the retail customers. Future research should focus on quantifying retail credit risk with respect to climate change effects. The following research questions are therefore proposed:

- What is the effect on credit before and after a climate event for retail customers?
- What is the effect in different sectors?

**Author Contributions:** N.M. contributed 70% of the total artilce preparation. Responsibilities include conceptualization, methodology, resources, data curation, writing, preparation, review and editing. H.T. supervised the whole process, and reviewed the work. All authors have read and agreed to the published version of the manuscript.

**Funding:** This research received no external funding.

**Informed Consent Statement:** Not applicable.

**Data Availability Statement:** The data will be made available upon request.

**Acknowledgments:** The authors wish to acknowledge the Center of Applied Data Sciences at University of Johannesburg and its staff for the support and advises on this project.

**Conflicts of Interest:** The authors declare no conflict of interest.

## Abbreviations

The following abbreviations are used in this manuscript:

| | |
|---|---|
| WoS | Web of Science |
| CFC | Chlorofluorocarbons |
| USA | United States of America |
| UK | United Kingdom |
| UN | United Nations |
| MFI | Microfinance Institution |
| CDP | Carbon Disclosure Project |

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
