# Peer review of "Quantifying Causality between Climate Change and Credit Risk: A Bibliometric Study and Research Agenda"

_sustainability, doi:10.3390/su15129319_

Round 1
Reviewer 1 Report
Ø To be legible, the whole text must be completely edited with the help of a native English editor to polish your writing.
Ø What has been studied Introduction should be clearly stated research questions and targets first. Then answer several questions: Why is the topic important (or why do you study on it)? What are research questions or objectives? What are your contributions? Why is to propose this particular method (This must come from Literature discussion)?
Ø Revised your introduction. Check the first paragraph of the introduction what’s its mean? From line 15-21.
Ø Research on climate change and credit risk has not been done extensively in the 40 past decades. It is only recently (2015) that we can now see an interest or spike on the 41 topic. The sudden interest was inspired by the 2015 Paris Agreement, where 195 countries 42 pledged to limit global warming levels by year 2050 [5] ( are you sure about that ? the research on climate change and credit risk has not been done last 40 decades?????? Line 40-43.
Ø The contribution is not well-positioned as compared to the existing literature. First, the scope of the literature review is not clear. Second, the contribution as compared to the existing literature is not well stated. What is really new in the research? Please underscore the scientific value added/contributions of your paper in your abstract and introduction and address your debate shortly in the abstract.
Ø The literature review is necessary for you to clarify the “contribution” of your study. Each reference mentioned should be discussed.
Ø In research Methods you have to utilize R software, explained why you used it? Why this software is useful for this study and data set?
Ø In Table 3 you have mentioned the Top 20 Institutions by number of articles also add the countries name with separate column.
Ø In table 6. Show the summary of each paper’s key focus and findings also add the country name with separate column.
Ø I would suggest you enhance your discussion 1. Policy implications and 2. Theoretical implications. 3. Please make sure your conclusions' section underscore the scientific value added of your paper, and/or the applicability of your findings/results, as indicated previously. Basically, you should enhance your findings, limitations, underscore the scientific value added of your paper, and/or applicability of your contributions/shortages and future study in this session.
Ø To be legible, the whole text must be completely edited with the help of a native English editor to polish your writing.
Ø What has been studied Introduction should be clearly stated research questions and targets first. Then answer several questions: Why is the topic important (or why do you study on it)? What are research questions or objectives? What are your contributions? Why is to propose this particular method (This must come from Literature discussion)?
Ø Revised your introduction. Check the first paragraph of the introduction what’s its mean? From line 15-21.
Ø Research on climate change and credit risk has not been done extensively in the 40 past decades. It is only recently (2015) that we can now see an interest or spike on the 41 topic. The sudden interest was inspired by the 2015 Paris Agreement, where 195 countries 42 pledged to limit global warming levels by year 2050 [5] ( are you sure about that ? the research on climate change and credit risk has not been done last 40 decades?????? Line 40-43.
Ø The contribution is not well-positioned as compared to the existing literature. First, the scope of the literature review is not clear. Second, the contribution as compared to the existing literature is not well stated. What is really new in the research? Please underscore the scientific value added/contributions of your paper in your abstract and introduction and address your debate shortly in the abstract.
Ø The literature review is necessary for you to clarify the “contribution” of your study. Each reference mentioned should be discussed.
Ø In research Methods you have to utilize R software, explained why you used it? Why this software is useful for this study and data set?
Ø In Table 3 you have mentioned the Top 20 Institutions by number of articles also add the countries name with separate column.
Ø In table 6. Show the summary of each paper’s key focus and findings also add the country name with separate column.
Ø I would suggest you enhance your discussion 1. Policy implications and 2. Theoretical implications. 3. Please make sure your conclusions' section underscore the scientific value added of your paper, and/or the applicability of your findings/results, as indicated previously. Basically, you should enhance your findings, limitations, underscore the scientific value added of your paper, and/or applicability of your contributions/shortages and future study in this session.
Author Response
I have attached the comments to the reviewers Notes.

Reviewer 2 Report
A well-researched and cogently written article with much attention to detail and carefully proofread. Please see additional comments in the attached file

A well-researched and cogently written article with much attention to detail and carefully proofread. Very few grammatical or spelling errors
Author Response
I have attached the comments to the reviewer's notes.

Reviewer 3 Report
See attachment.

Round 2
Reviewer 1 Report
Accepted
Author Response
Thanks. All is in order.
Reviewer 3 Report
I consider that the following questions/suggestions were not resolved:
(2) the aim is to analyse the causality between climate change and credit risk; the study does not examine the proxies that measure climate change and credit risk. I propose to consider this aspect of research. (3) Section 3.2 should be the most crucial section of the article. A deeper analysis of this issue should be considered, and this would be of more substantial benefit to future researchers.
I do not have additional suggestions.
Author Response
1) Proxies: The previous studies used the following variables to quantify climate related risks: remotely-sensed vegetation health indices which are vegetation condition index, temperature condition index, vegetation health index to explain climate risk. Other studies employed carbon emissions like floods , extreme weather and extreme temperatures ,climate change score. Credit risk is widely quantified by using the measures like distance to default, loan (debt) amount, mortgage refinancing, credit ratings, probability of default, and credit scoring.
2) Please see section 3.2, it has been updated.